# An old drug and different ways to treat cutaneous leishmaniasis: Intralesional and intramuscular meglumine antimoniate in a reference center, Rio de Janeiro, Brazil

**Carla Oliveira-Ribeiro**[1,2©]*, **Maria Inês Fernandes Pimentel**[1©], **Liliane de Fátima Antonio Oliveira**[1©], **Érica de Camargo Ferreira e Vasconcellos**[3,4©], **Fatima Conceição-Silva**[5‡], **Armando de Oliveira Schubach**[1©], **Aline Fagundes**[1‡], **Cintia Xavier de Mello**[6‡], **Eliame Mouta-Confort**[1‡], **Luciana de Freitas Campos Miranda**[1‡], **Claudia Maria Valete-Rosalino**[1,7‡], **Ana Cristina da Costa Martins**[1‡], **Raquel de Vasconcellos Carvalhaes de Oliveira**[8©], **Leonardo Pereira Quintella**[9‡], **Marcelo Rosandiski Lyra**[1©]

1 Laboratório de Pesquisa Clínica e Vigilância em Leishmanioses, Instituto Nacional de Infectologia Evandro Chagas, Fundação Oswaldo Cruz, Rio de Janeiro, Brazil, 2 Centro de Transplante de Medula Óssea, Instituto Nacional de Câncer José Alencar Gomes da Silva, Rio de Janeiro, Brazil, 3 Escola de Medicina Souza Marques, Rio de Janeiro, Rio de Janeiro, Brazil, 4 Faculdade de Medicina, Centro Universitário de Volta Redonda, Rio de Janeiro, Brazil, 5 Laboratório de Imunoparasitologia, Instituto Oswaldo Cruz, Fundação Oswaldo Cruz, Rio de Janeiro, Brazil, 6 Laboratório Interdisciplinar de Pesquisas Médicas, Instituto Oswaldo Cruz, Fundação Oswaldo Cruz, Rio de Janeiro, Brazil, 7 Departamento de Otorrinolaringologia e Oftalmologia, Faculdade de Medicina, Universidade Federal do Rio de Janeiro, Rio de Janeiro, Brazil, 8 Laboratório de Epidemiologia Clínica, Instituto Nacional de Infectologia Evandro Chagas, Fundação Oswaldo Cruz, Rio de Janeiro, Brazil, 9 Serviço de Anatomia Patológica, Instituto Nacional de Infectologia Evandro Chagas, Fundação Oswaldo Cruz, Rio de Janeiro, Brazil

© These authors contributed equally to this work.
‡ These authors also contributed equally to this work.
* carla_ribb@yahoo.com.br

**Data Availability Statement:** The dataset of the study is available at Fiocruz's institutional data

## Abstract

### Background

Treatment of cutaneous leishmaniasis (CL) remains challenging since the drugs currently used are quite toxic, thus contributing to lethality unrelated to the disease itself but to adverse events (AE). The main objective was to evaluate different treatment regimens with meglumine antimoniate (MA), in a reference center in Rio de Janeiro, Brazil.

### Methodology

A historical cohort of 592 patients that underwent physical and laboratory examination were enrolled between 2000 and 2017. The outcome measures of effectiveness were epithelialization and complete healing of cutaneous lesions. AE were graded using a standardized scale. Three groups were evaluated: Standard regimen (SR): intramuscular (IM) MA 10–20 mg $Sb^{5+}$/kg/day during 20 days (n = 46); Alternative regimen (AR): IM MA 5 mg $Sb^{5+}$/kg/day during 30 days (n = 456); Intralesional route (IL): MA infiltration in the lesion(s) through subcutaneous injections (n = 90). Statistical analysis was performed through Fisher exact and Pearson Chi-square tests, Kruskal-Wallis, Kaplan-Meier and log-rank tests.

repository (https://dadosdepesquisa.fiocruz.br), under the DOI https://doi.org/10.35078/ZVATUJ. How to cite the dataset: Lyra, Marcelo Rosandiski; Pimentel, Maria Inês Fernandes; Ribeiro, Carla de Oliveira, 2021, "An old drug and different ways to treat cutaneous leishmaniasis: intralesional and intramuscular meglumine antimoniate in a reference center, Rio de Janeiro, Brazil.", https://doi.org/10.35078/ZVATUJ, Oswaldo Cruz Foundation, V1. Requests to access to the dataset can be made through the following e-mail: maria.pimentel@ini.fiocruz.br After analysis of the dataset owner, a private link can be created to give access to the requesting researcher who has his request approved by the owner. Access to the requesting researcher will depend on an analysis of the type of access request. Some conditions are crucial for the availability of data use (authors citation and non-commercial use, for example). Access will be available according to the "Lei Geral de Proteção de Dados" (General Law on Data Protection), Law Number 13,709, approved in August 2018 and effective from August 2020 in Brazil.

**Funding:** This study was funded by Coordenação de Pessoal de Nível Superior (CAPES) [grant number 001]. It was also funded by Conselho Nacional de Desenvolvimento Científico e Tecnológico (CNPq) [grant numbers AOS 304335/2014-2, CMVR 313327/2018-1]. This study was partially supported by the Coordination for the Improvement of Higher Education Personnel (Coordenação de Aperfeiçoamento de Pessoal de Nível Superior - CAPES) - Finance Code 001. The funders had no role in study design, data collection and analysis, decision to publish, or preparation of the manuscript.

**Competing interests:** The authors have declared that no competing interests exist.

## Results

SR, AR and IL showed efficacy of 95.3%, 84.3% and 75.9%, with abandonment rate of 6.5%, 2.4% and 3.4%, respectively. IL patients had more comorbidities (58.9%; p = 0.001), were mostly over 50 years of age (55.6%), and had an evolution time longer than 2 months (65.6%; p = 0.02). Time for epithelialization and complete healing were similar in IL and IM MA groups (p = 0.9 and p = 0.5; respectively). Total AE and moderate to severe AE that frequently led to treatment interruption were more common in SR group, while AR and IL showed less toxicity.

## Conclusions/Significance

AR and IL showed less toxicity and may be good options especially in CL cases with comorbidities, although SR treatment was more effective. IL treatment was an effective and safe strategy, and it may be used as first therapy option as well as a rescue scheme in patients initially treated with other drugs.

## Author summary

Treatment of cutaneous leishmaniasis remains a challenge since the drugs used are quite toxic. Currently, there is a global effort to reduce the morbidity associated with the treatment of this disease and life-threatening complications due to drugs or treatment approaches. Meglumine antimoniate (MA) in different regimens was evaluated in cutaneous leishmaniasis patients in the state of Rio de Janeiro, Brazil. Effectiveness and toxicity were compared among the groups: standard regimen (SR) [intramuscular (IM) MA in the dosage of 10 to 20 mg of pentavalent antimony ($Sb5^+$)/kg/day]; alternative regimen (AR) [IM MA in the dosage of 5 mg $Sb5^+$/kg/day]; and intralesional route (IL) [patients treated with MA through the infiltration of the lesion]. AR and IL regimens demonstrated good effectiveness, with reduced abandonment rate and toxicity. Total adverse events were higher in the SR group, which frequently led to treatment interruptions. AR and IL showed less toxicity especially in CL cases with comorbidities, although SR treatment was more effective than AR and IL regimens. IL was an effective and safe treatment and may be used as a first therapy option as well as a rescue scheme.

## Introduction

Cutaneous leishmaniasis (CL) in the New World is a disease with low lethality, but its treatment with pentavalent antimonials frequently involves serious and potentially lethal side effects [1,2,3,4,5]. Alternative treatments to intravenous or intramuscular meglumine antimoniate (MA) have thus been proposed. Among them, intralesional (IL) infiltration appears to be an effective and easily reproducible option [6,7].

In 2017, the Brazilian Ministry of Health (BMH) recommended for the first time the technique of IL infiltration of MA as a therapeutic option in patients with localized CL with a single lesion up to 3.0 centimetres (cm) in its largest diameter, in any location except head and periarticular areas [8]. The same guide suggested an alternative low dosage regimen of 5 mg of pentavalent antimony ($Sb^{5+}$) / kg / day [alternative regimen (AR)], in addition to the classic dosage of 10–20 mg of $Sb^{5+}$ / kg /day [standard regimen (SR)] [8].

Leishmaniasis Clinical Research and Surveillance Laboratory at Evandro Chagas National Institute of Infectious Diseases, Oswaldo Cruz Foundation (INI / Fiocruz) is a reference center for the treatment of American tegumentary leishmaniasis (ATL) in the state of Rio de Janeiro, with experienced professionals in the diagnosis and the management of the disease. For more than three decades, AR and IL MA have been used at this center when intravenous or intramuscular treatments are formally contraindicated [9,10,11,12,13]. Intramuscular treatment with AR has been widely used in the referral center with good effectiveness in patients from all over the country [14], and lower toxicity when compared to SR. Even in cases with formal contraindication for intravenous/intramuscular route (increased liver enzymes, uncontrolled *diabetes mellitus* or systemic arterial hypertension, heart disease, the elderly, among others), AR was well tolerated [14,15,16].

We aimed to evaluate the effectiveness and toxicity among different treatment regimens with MA (SR, AR and IL) used at this reference center between 2000 and 2017. Our hypothesis was that treatments with alternative and IL MA regimens had reasonable effectiveness, with the advantage of reduced toxicity when compared to SR.

## Methods

### Ethics statement

This study was approved by the Ethics Committee of INI/Fiocruz (number 81787917.1.0000.5262). All participants signed an informed consent form. Patients under 18 had the informed consent form signed by parents or guardians who accompanied them during all the diagnostic and therapeutic procedures.

### Groups and monitoring parameters

A retrospective cohort study of 592 patients treated at INI/Fiocruz with MA (via intramuscular or IL) as the first therapeutic option between January 2000 to December 2017 was conducted. All patients had parasitological diagnostic confirmation through one or more of the following exams: direct exams (scraping or imprint), histopathology, isolation in culture with Novy-MacNeal-Nicolle (NNN) biphasic medium plus Schneider's Drosophila Medium (Sigma-Aldrich, St. Louis, Missouri, USA) according to protocol registered in https://dx.doi.org/10.17504/protocols.io.22tggen, or polymerase chain reaction (PCR) of material/tissue collected from the cutaneous lesions. This study included only CL treatment-naive patients.

### Etiological identification of *Leishmania* species

When possible, parasitic isolation in culture was performed by the techniques: Multilocus enzyme electrophoresis (MLEE), PCR, Restriction fragment length polymorphism (RFLP) and Sequencing.

**Multilocus enzyme electrophoresis (MLEE).** The MLEE analysis was performed on 394 samples. The total culture volume was centrifuged and the pellet was submitted to three washes in NaCl-EDTA buffer under centrifugation to obtain the parasite mass for MLEE analysis. MLEE was performed on 1% agarose gel supported by GE Healthcare CalBond (124 X 258 mm), using six or seven enzymatic systems: GPGDH (6-phosphogluconate dehydronegase, EC 1.1.1.43); GPI (glucose phosphate isomerase, EC 5.3.1.9); NH (nucleotidase, EC 3.2.2.1); G6PDH (glucose 6-phosphate dehydrogenase, EC 1.1.1.49); PGM (phosphoglucomutase, EC 5.4.2.2); ME (malic enzyme, EC 1.1.1.40) or MDH (malate dehydrogenase, EC 1.1.1.37). The MDH enzymatic system was used only when variant profiles were detected in the other enzymatic systems. Isoenzyme electrophoresis was performed with the reference strain of

*Leishmania (Viannia) braziliensis* (MHOM/BR/M2903). If any sample presented a different profile, a new assay was performed with the other reference strains: *Leishmania (Leishmania) amazonensis* (IFLA/BR/19767/PH8), *L. (V.) guyanensis* (MHOM/BR/1975/M4147), *L. (V.) shawi* (MCED/BR/1984/M8408), *L. (V.) lainsoni* (MHOM/BR/1981/M6426), *L. (V.) naiffi* (MDAS/BR/1979/M5533), and *L. (L.) infantum* (MHOM/BR/1974/PP75). Analysis of gel bands was performed qualitatively, by visual comparison of the sample band with the default reference strains.

**Polymerase chain reaction (PCR).**   Seven samples with reduced growth in the isolation in culture and three that presented variant profiles in one or more enzymatic systems using the MLEE technique were also analyzed by PCR-RFLP, for species confirmation. DNA extraction was performed using a DNAzol Reagent kit (Invitrogen, Thermo Fisher Scientific, Waltham, Massachusetts, USA), following the manufacturer's recommendations. PCR assays were performed using the primers 5'GGACGAGATCGAGCGCATGGT3' and 5'TCCTTCGACGCC TCCTGGTTG3', to amplify a 234-bp fragment of the gene region encoding hsp70C. The amplification products were separated using 2% agarose gel electrophoresis with ethidium bromide (0.5 μg/mL) and visualized under ultraviolet light.

**Restriction fragment length polymorphism (RFLP).**   Amplification products obtained by PCR were digested with two restriction enzymes, *Hae*III (Sigma-Aldrich, Saint Louis, Missouri, USA) and *Bst*UI (Thermo Fisher Scientific, Waltham, Massachusetts, USA), following the manufacturer's recommendations. The fragments obtained by enzymatic digestion were separated on a 12.5% polyacrylamide gel and stained with silver, and bands were compared with a DNA fragment size marker (100-bp DNA ladder). The banding pattern was compared with the reference strains: *L. (V.) braziliensis* (MHOM/BR/1975/M2903), *L. (L.) amazonensis* (IFLA/BR/1967/PH8), *L. (V.) guyanensis* (MHOM/BR/1975/M4147), *L. (V.) shawi* (MCEB/ BR/1984/M8408), *L. (V.) lainsoni* (MHOM/BR/1981/M6426), and *L. (V.) naiffi* (MDAS/BR/ 1979/M5533).

**Sequencing.**   Sanger sequencing of the internal transcribed spacer of ribosomal DNA (ITS1-rDNA) was performed only in one case where it was not possible to obtain taxonomic identification through MLEE or PCR-RFLP.

The ITS1-rDNA was amplified by conventional PCR using the primers L5.8S: 50-TGATA CCACTTATCGCACTT-30 and LITSR: 50-CTGGATCATTTTCCGATG-30. Amplification reaction was performed in volume of 50 μL. Amplicons from the PCR positive sample were visualized on 2% agarose gel and purified using the Wizard SV Gel kit and PCR Clean-up System kit (Promega, Madison, Wisconsin, USA). The products were then sequenced with the same primers used in the PCR assay. Sequencing was performed on an automated sequencer at *Plataforma de Sequenciamento Genômico* ABI-3730 (Oswaldo Cruz Institute/Fiocruz).

Sequence alignment was performed using SeqMan Pro (DNASTAR, Madison, Wisconsin, USA) and comparisons were conducted with *Leishmania* reference strains sequences obtained from the GenBank database. Phylogenetics analyses with the evolutionary history were inferred using the maximum likelihood method based on the Jukes-Cantor model and the sequences were aligned using Molecular Evolutionary Genetic Analysis (MEGA) version 6 (Tokyo Metropolitan University, Tokyo, Japan; Arizona State University, Arizona, USA; King Abdulaziz University, Jeddah, Saudi Arabia).

## Exclusion criteria

Reasons for excluding subjects were the following: absence of parasitological confirmation; cases of mucosal, mucocutaneous, diffuse or disseminated leishmaniasis; patients with spontaneous healing of cutaneous lesions; patients who were diagnosed at INI/Fiocruz and were not

treated there or who were followed-up after treatment in other health units; patients with diseases that altered the immune system; patients in whom no informed consent form was obtained. Patients who were initially treated with other drugs different from MA were also excluded.

## Follow-up of the patients

Patients were followed up with physical examination and laboratory tests according to the protocol used at INI / Fiocruz before, every seven to ten days during and soon after treatment [16]. Questioning about the use of non-specific medications for the treatment of other comorbidities (such as *diabetes mellitus*, systemic arterial hypertension, among others) was carried out in the first evaluation. Lesions, ulcers and scars were evaluated by dermatologists and standardized photographs were taken every two weeks after treatment for two months; then every month for three months; then every three months until the end of one year; then every six months until the end of two years; and then once a year [16]. Subjects were further assessed to rule out mucosal lesions, through video fiberoptic nasolaryngoscopy, performed by an Ear, Nose and Throat (ENT) specialist in the first medical visit; when epithelialization of the ulcers occurred; then every two to three months until the end of one year; then every six months until the end of two years; then once a year [16].

Although there were variations in the number of IL infiltrations performed, the time between sessions was approximately 15 days, according to institutional protocols. When there was an interruption due to adverse events (AE), they were signaled. The interruption of treatment due to clinical cure occurred when epithelialization and progression to complete healing of the lesion were verified in the clinical exam.

AE were graded using an adapted toxicity scale from the Division of AIDS Table for Grading of Severity of Adult and Pediatric Adverse Events [17,18].

Three groups were evaluated:

Standard regimen (SR): patients treated with systemic MA in the dosage of 10–20 mg of pentavalent antimony ($Sb^{5+}$)/kg/day for 20 days, as recommended by Brazilian guidelines [8].

Alternative regimen (AR): patients treated with systemic MA in the dosage of 5 mg $Sb^{5+}$/kg/day for 30 days. As a general rule, patients with CL are treated with AR at INI/Fiocruz by clinical choice [14,15,16].

Intralesional route (IL): patients treated with MA by the infiltration of the drug within the skin lesion through subcutaneous injections, according to a previously described technique. Briefly, the drug was injected subcutaneously, using a long medium-calibre needle to inject the volume necessary to infiltrate the base of the lesion until raising it and producing intumescence [6]. In general, patients with formal contraindication to intravenous/intramuscular therapy are treated with IL MA at INI/Fiocruz.

The maximum daily dose for systemic treatment for adults is 3 ampoules of 5.0 mL (or 1.215mg) $Sb^{5+}$/kg/day as recommended by the BMH (2017). For IL treatment, this maximum daily limit for the use of the drug was not applicable.

## Definitions

Case of CL: patient with a cutaneous lesion(s) and parasitological confirmation of the disease that initiated treatment with MA, without mucous lesions.

Primary outcome: clinical cure: defined as complete epithelialization of all lesions. Epithelialization was defined as complete wound closure without erosions or crusts until the appointment on the 120th day from the beginning of the treatment (recent cure) [16].

Secondary outcome: definite scar, defined as the presence of complete healing of all lesions (complete epithelialization and absence of crusts, infiltration or desquamation) [16] at day 180 from the beginning of the treatment [19].

No cure: inability to achieve clinical cure with epithelialization and definite scar [16].

Relapse: the reappearance of an active lesion after complete cure (definite scar) or development of new lesions in other locations after scar was established [16].

Abandonment: when there was no evidence of clinical cure and the patient did not attend a meeting for 30 days after the third scheduling for evaluation [8]; or incomplete therapeutic regimen.

## Measurement of lesion(s)

All lesions of the evaluated patients were measured in their largest and smallest diameters. The area of the lesion was calculated according to the ellipse formula (smaller radiusxgreater radiusxπ). We measured the lesion diameter in milimetres (mm) at the external margins of the lesion border using a transparent ruler. Most of the cases presented single lesions. However, in cases with three or more lesions, we considered the measurements and topography of the largest lesions present at the initial clinical examination.

## Variables

Sociodemographic variables (sex, age and likely infection site); use of non-specific medications; data on the past medical history including clinical comorbidities (systemic arterial hypertension, *diabetes mellitus*, heart disease, dyslipidemia, among others); characteristics of the lesions (number, duration, size, morphology and topography); results of laboratory tests; were collected from medical records.

## Statistical analysis

The frequency of qualitative variables and summary measures (median, minimum and maximum) for continuous variables were used. The cure rates were presented as percentages with their respective 95% confidence intervals (CI).

The association of qualitative variables according to treatment (SR, AR and IL) was evaluated through Fisher's Exact and Pearson Chi-square test. The Kolmogorov-Smirnov test indicated rejection of the normality of the continuous variables, and hence the comparison of continuous variables according to treatment (SR, AR and IL) was performed using the Kruskal-Wallis test. The area of the lesions greater than 900 $mm^2$, concerning epithelialization and complete healing, according to treatment groups was analyzed by Pearson Chi-square test.

Considering that the main objective was to compare the cure proportion among the different treatments, we used the 5% significance level, the proportions of cure and sample sizes of each pair of groups. We found the following estimations for the study power: SR/AR = 39.7%; SR/IL = 78.1%; AR/IL = 43.6%.

We used survival analysis to investigate differences among the groups, considering the time until the occurrence of each outcome (epithelialization and complete healing of cutaneous lesions). For epithelialization, the censors were cases that did not epithelialize or those who abandoned the follow-up without presenting an epithelialized lesion. Censoring according to complete healing outcome was defined by cases that did not heal or those who abandoned follow-up without complete healing. Censoring was carried out on the date of the last evaluation.

Kaplan-Meier method estimated the survival probability stratified according to qualitative variables (sex, age, likely infection site, comorbidities, number, duration, size, morphology, topography of the lesions and results of laboratory tests). The log-rank test was used to check for differences between the stratified survival curves. P-values< 0.05 indicated statistically

significant tests. Data were analyzed by the software R version 4.0.0 from the R Foundation for Statistical Computing Platform, Vienna, Austria.

## Results

### Sample

Fig 1 shows the flowchart for inclusion in the study.

### Sociodemographic variables

Five hundred and ninety-two patients were analyzed. The median age was 35 (1–93) years. Sixty-two per cent (n = 371) were men.

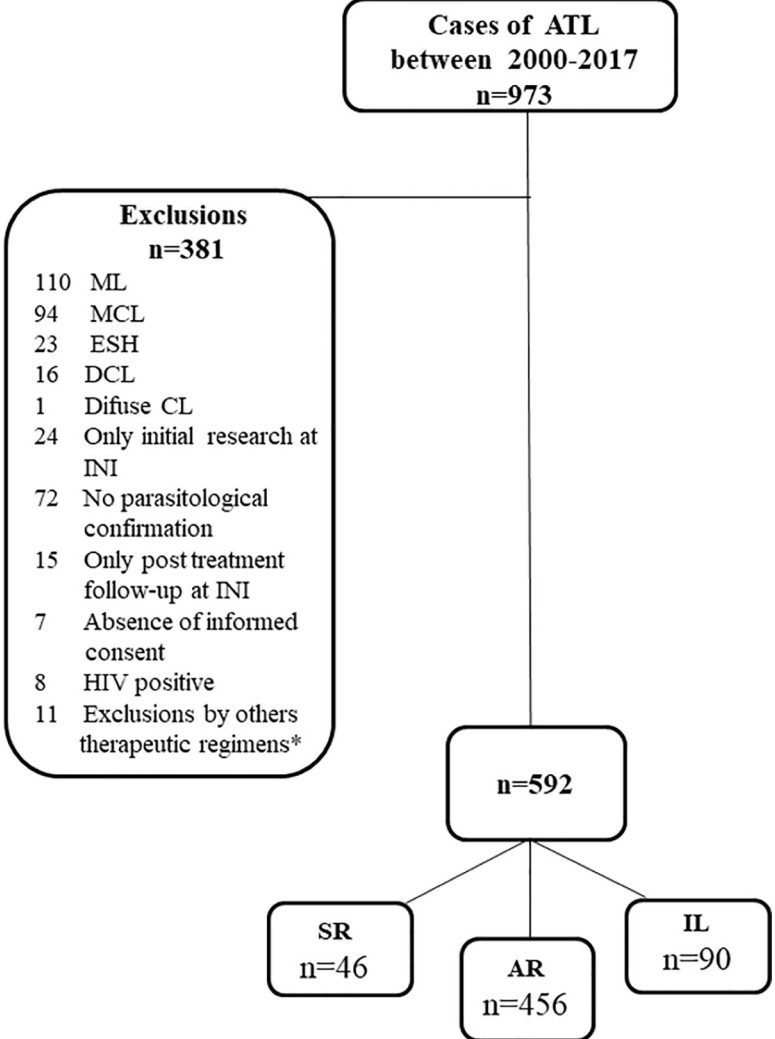

**Fig 1. Flowchart of inclusion and exclusion of the studied patients, for the population of patients with American tegumentary leishmaniasis treated at INI, 2000–2017.** MA- meglumine antimoniate; INI- Evandro Chagas National Institute of Infectious Diseases; Fiocruz- Oswaldo Cruz Foundation; ML- mucosal leishmaniasis; MCL- mucocutaneous leishmaniasis; ESH- early spontaneous resolution without treatment; DCL- disseminated cutaneous leishmaniasis; CL- cutaneous leishmaniasis; HIV- human immunodeficiency virus; LCL- localized cutaneous leishmaniasis; SR- standard regimen with MA 10 to 20 mg/kg/day; AR- alternative regimen with MA 5 mg/kg/day; IL- intralesional MA. *MA used as first therapeutic regimen.

The characteristics of the patients according to the treatments used are shown in Table 1. There was a predominance of patients from the state of Rio de Janeiro (93.2%; p = 0.001). There were no statistically significant differences between patients from Rio de Janeiro and other states regarding the epithelialization of the lesions (p = 0.798).

IL group had more comorbidities, the majority were over 50 years, and the use of non-specific medications was more common in this group, Table 1.

## Characteristics of treated lesions

The predominant clinical presentation was a single (n = 394; 66.5%) and ulcerated lesion (n = 495; 83.6%). Considering the whole cohort, the median of the largest diameter of the lesions was 30.0 mm (4–130). Approximately 17% of patients had 2 lesions on the first evaluation [(median of the largest diameter of the larger lesion was 30.0 mm (5–80), and of the smaller lesion was 20.0 mm (2–70)]. 16% of patients had 3 or more lesions on the first evaluation [(median of the largest diameter of the largest lesion was 25.0 mm (5–100), of the second largest lesion was 20.0 mm (3–90), and of the third largest lesion was 20.0 mm (3–85)].

The median time of progression of the lesion (evolution time) prior to diagnosis was 60 days. In 65.6% of the cases in the IL group, this time was longer than 2 months (Table 1).

One hundred fifty-one treated lesions had an area greater than 900 mm$^2$. There were no significant differences among the groups regarding the area of the lesions greater than 900 mm$^2$ concerning epithelialization and complete healing (p = 0.251; p = 0.061, respectively). Eighty lesions were located in the cephalic segment or periarticular regions. There were no statistically significant differences regarding the epithelialization and healing of the lesions and their location in the above-mentioned topographies when compared to other locations (p = 0.08; p = 0.51, respectively). Patients in the AR group presented 64 lesions, those in the SR group presented 8, and those in the IL group presented with 7 lesions that were located in the cephalic segment or periarticular regions. There were no statistically significant differences among the groups regarding epithelialization and complete healing of lesions located in those particular topographies (p = 0.945; p = 0.256, respectively).

## IL group

Sixty-five treated patients had a single lesion with a median area of 748.9 mm$^2$ (19.6–4670.7). In 14 cases, two treated lesions had a median area of 269.62 mm$^2$ (15.7–1511.3). Eleven patients had three or more lesions with a median area of 317.23 mm$^2$ (11.8–900.4). The median number of infiltrations was two (1–6), with a median of 8.0 mL/session (1–30) and the median time of 15.6 days between infiltrations. Among the patients in the IL group, seven (7.8%) were treated with a dosage of AM IL greater than 15 mL/session. None of these cases had AE.

## Other analyzed variables

There were no statistically significant differences among the groups regarding sex and the characteristics of the lesions (number, larger diameter, smaller diameter, location, time of evolution before diagnosis), positivity in the culture, visualization of amastigotes in the histopathological exam, and follow-up period (Table 1).

## Outcomes after first treatment

In the SR group, we obtained a complete healing rate of 95.3% (41/43, CI: 84.2–99.4); 30.4% of treatment interruption; and 6.5% of abandonment rate (Fig 2).

**Table 1. Distribution of the clinical and laboratory characteristics of 592 patients according to group treatment.**

| Variables | IL | AR | SR | p-value[a,b] |
|---|---|---|---|---|
| | n = 90 (%) | n = 456 (%) | n = 46 (%) | |
| **Sex** (n = 592) | | | | 0.247 |
| Female | 30 (33.3) | 178 (39.1) | 13 (28.3) | |
| Male | 60 (66.7) | 278 (60.9) | 33 (71.7) | |
| **Likely location of infection** (n = 592) | | | | **0.001** |
| Rio de Janeiro state | 71 (78.9) | 436 (95.6) | 45 (97.8) | |
| Other Brazilian states | 19 (21.1) | 20 (4.4) | 1 (2.2) | |
| **Number of lesion(s)** (n = 590) | | | | |
| One lesion | 65 (72.2) | 295 (64.8) | 34 (73.9) | 0.680 |
| Two or more lesions | 25 (27.8) | 159 (35.0) | 12 (26.1) | |
| **Age** (n = 592) | | | | **0.001** |
| <50 years | 40 (44.4) | 349 (76.5) | 33 (71.7) | |
| ≥ 50 years | 50 (55.6) | 107 (23.5) | 13 (28.3) | |
| **Lesion 1 largest diameter (mm)** (n = 541) | | | | 0.583 |
| < 30 mm | 44 (53.0) | 233 (56.4) | 22 (48.9) | |
| ≥ 30 mm | 39 (47.0) | 180 (43.6) | 23 (51.1) | |
| **Lesion 2 largest diameter (mm)** (n = 178) | | | | 0.823 |
| < 50 mm | 21 (95.5) | 131 (91.6) | 12 (92.3) | |
| ≥ 50 mm | 1 (0.5) | 12 (8.4) | 1 (7.7) | |
| **Comorbidity** (n = 491) | | | | |
| Yes | 53 (79.1) | 217 (55.1) | 20 (66.7) | **0.001** |
| No | 14 (20.9) | 177 (44.9) | 10 (33.3) | |
| **Use of non-specific medication** (n = 475) | | | | **0.001** |
| Yes | 35 (61.4) | 115 (30.4) | 14 (46.7) | |
| No | 22 (38,6) | 273 (69.6) | 16 (53.3) | |
| **Location of lesion(s)** (n = 589) | | | | |
| Head / Neck | 11 (12.2) | 86 (18.9) | 10 (21.7) | 0.129 |
| Trunk | 9 (10.0) | 58 (12.8) | 2 (4.3) | |
| Upper limbs | 40 (44.4) | 137 (30.1) | 14 (30.4) | |
| Thigh | 8 (8.9) | 31 (6.8) | 1 (2.2) | |
| Foot / Leg | 22 (24.4) | 141 (31.0) | 19 (41.3) | |
| **Amastigote in Histopathology** (n = 513) | | | | |
| Presence | 51 (66.2) | 246 (62.9) | 30 (71.4) | 0.430 |
| Absence | 29 (33.8) | 145 (37.1) | 12 (28.6) | |
| **Culture** (n = 522) | | | | 0.592 |
| Positive | 69 (89.6) | 360 (89.8) | 39 (88.6) | |
| Negative | 8 (10.9) | 41 (10.2) | 5 (11.4) | |
| **Evolution time of the lesion(s)** (n = 592) | | | | **0.02** |
| < 2 months | 31 (34.4) | 248 (54.5) | 23 (50.0) | |
| ≥ 2 months | 59 (65.6) | 208 (45.5) | 23 (50.0) | |
| | Median[min-max] | Median [min-max] | Median [min-max] | |
| **Age (years)** (n = 592) | 42 [4–84] | 33 [1–93] | 37 [15–72] | **0.001** |
| **Evolution time of the lesion(s) (months)** (n = 592) | 3 [1–72] | 2 [1–45] | 2.5 [1–48] | **0.007** |
| **Larger diameter of lesion(s) (mm)** (n = 541) | 30 [4–98] | 30 [5–130] | 32 [10–100] | 0.3 |
| **Number of lesions** (n = 590) | 1 [1–8] | 1 [1–7] | 1 [1–7] | 0.349 |

SR- standard regimen: 10–20 mg of $Sb^{5+}$/kg/day for 20 days; AR- alternative regimen: 5 mg $Sb^{5+}$/kg/day for 30 days; IL- intralesional route: drug infiltration in the skin lesion through subcutaneous injections.

[a] p-value < 0.05 indicates significant association in Pearson Chi-square or Kruskal Wallis test.

[b] Bold: significant p-values.

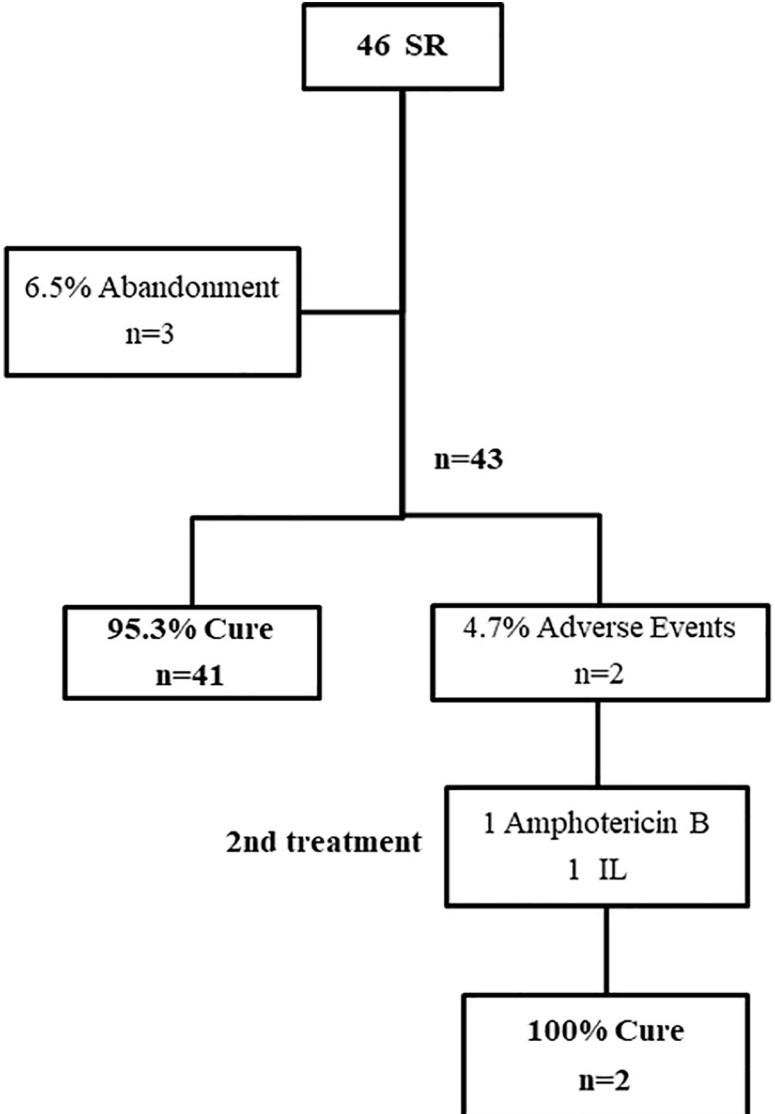

**Fig 2. Patients follow-up: standard regimen group.** SR- standard regimen; IL- intralesional route.

AR group presented complete healing rate of 84.3% (375/445, CI: 80.5–87.5); 3.1% of treatment interruption and 2.4% of abandonment rate (Fig 3).

The evolution with mucosal lesions after treatment occurred in two patients (0.3%) of the entire cohort (one patient from the IL group, one from the AR group). The median time for the appearance of these mucosal lesions was 131 days.

In 107 patients over the age of 50, the cure rate was 85% (n = 91).

Among the 40 patients who acquired the infection in other Brazilian states, 34 (85%) underwent clinical cure with the first treatment. Among the six patients who did not reach cure, three were additionally treated with MA (one with AR; two with IL), one with amphotericin B and two with pentamidine. Four patients healed after this second therapeutic course, there was one abandonment and the patient treated with AR relapsed. This patient was subsequently treated with amphotericin B and was then cured.

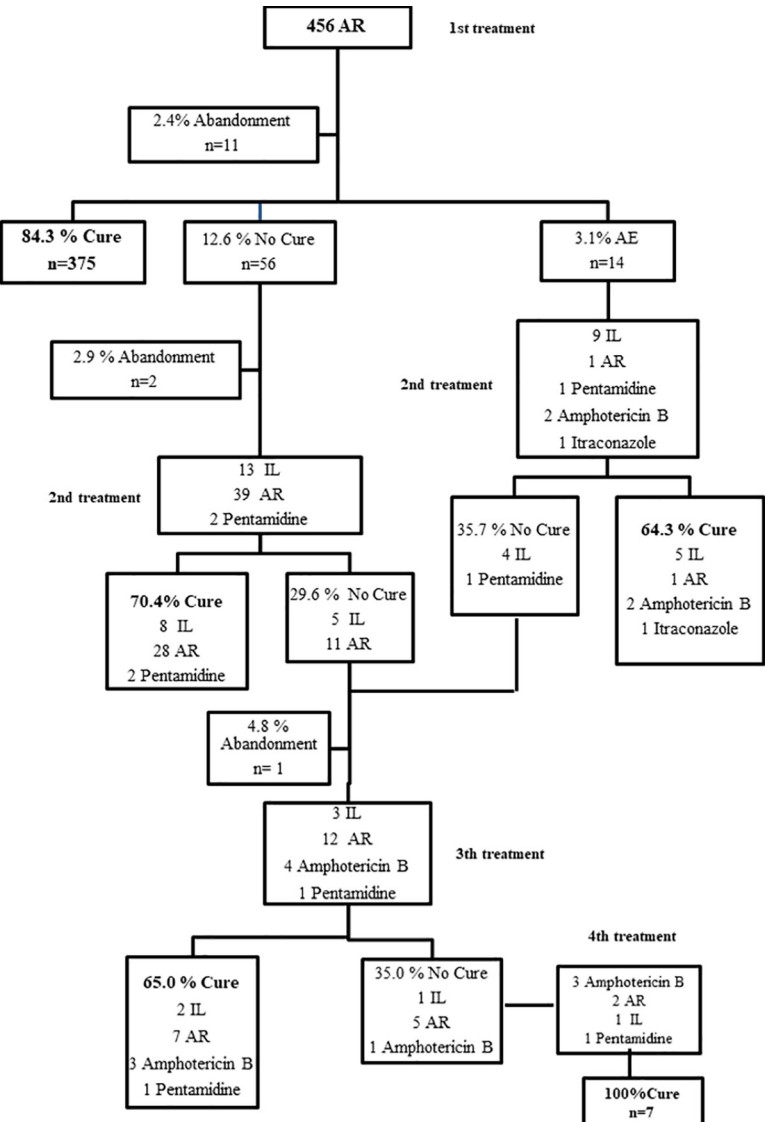

**Fig 3. Patients follow-up: alternative regimen group.** AR- alternative regimen; SR-standard regimen; IL-intralesional route; AE- adverse events.

In the IL group, treatment effectiveness (complete healing) was 75.9% (66/87, CI: 65.5–84.4) and an abandonment rate of 3.4% (n = 3) was noted (Fig 4).

No patient needed to interrupt treatment due to side effects.

In 50 patients over the age of 50, the cure rate was 74% (n = 37).

The SR group showed a higher complete healing rate than other groups regarding cure rate with the first course of treatment (p = 0.010). There were no statistical differences between AR and IL groups concerning the complete healing rate with the first therapeutic regimen (p = 0.4).

Table 2 summarizes the indications for IL treatment in patients treated at INI/Fiocruz. One hundred and four out of the 592 patients needed a second course of treatment, 30.8% of these cases received IL scheme as rescue therapy [median number of infiltrations of 2 (1–3), median of 6.5 mL/session (1–20)] with an efficacy rate of 71.8%.

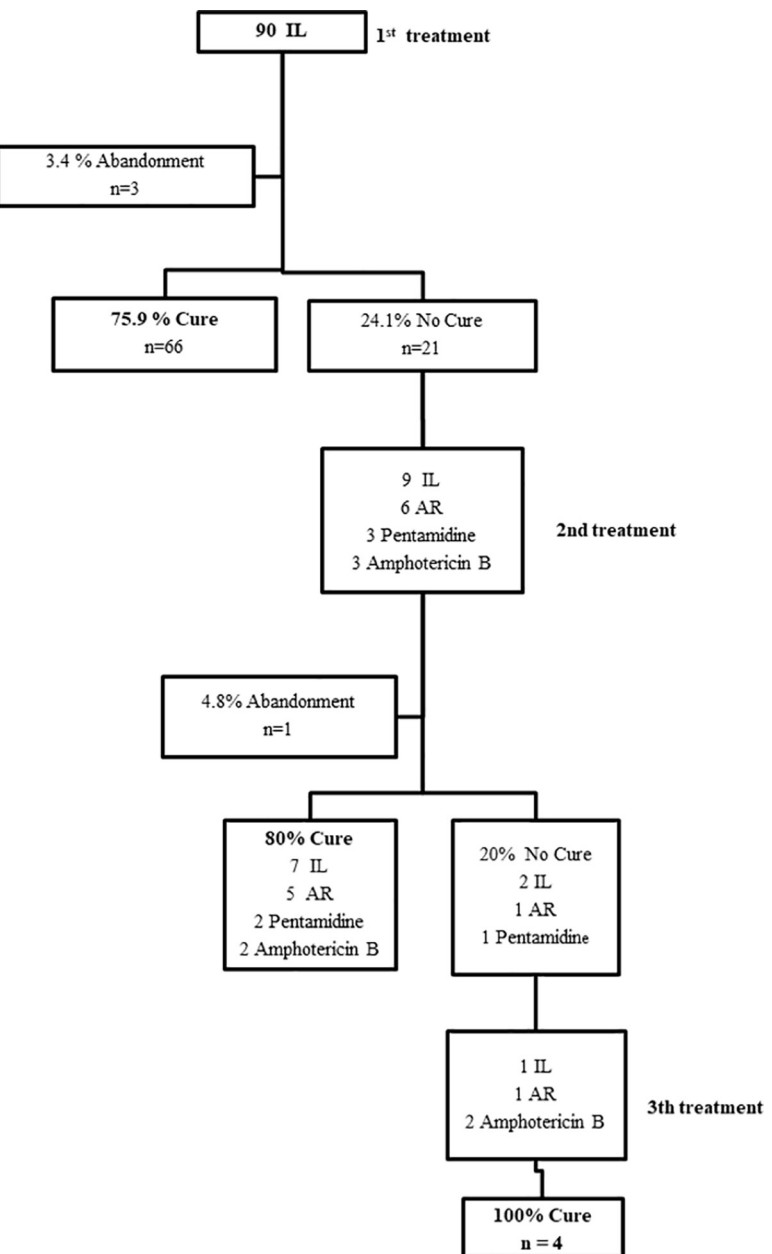

**Fig 4. Patients follow-up: intralesional route.** IL- intralesional route; AR- alternative regimen; SR- standard regimen.

### Results of survival analysis

The median follow-up period of the whole cohort was 995.6 (14–4478) days.

Epithelialization of the lesions occurred in 98% of the 494 evaluated patients (Fig 5A) and complete healing in 94% of the 498 evaluated patients (Fig 5B). We censored 10 patients for epithelization and 29 for complete healing analysis. The median time for epithelialization of the lesions was 40 days ranging from 5 to 228 days, and the median time for complete healing was 114 days ranging from 7 to 474 days.

Fig 5C shows that the time for epithelialization of the lesions is similar for the patients treated with IL MA compared to the patients treated with intramuscular MA (p = 0.9).

**Table 2. Indications for treatment with intralesional meglumine antimoniate.**

| IL | | IL | | IL | | IL | |
|---|---|---|---|---|---|---|---|
| 1st treatment | N = 90 | 2nd treatment | N = 32 | 3rd treatment | N = 4 | 4th treatment | N = 1 |
| Contraindication to systemic MA[a] | n = 47 | Failure of 1st treatment | n = 23 | Contraindication to systemic MA[c] | n = 2 | Failure of IM treatment | n = 1 |
| Patient´s choice | n = 43 | Adverse event in the 1st treatment[b] | n = 9 | Failure of IM treatment | n = 2 | | |

IL- intralesional route; MA- meglumine antimoniate; IM-intramuscular

[a] 11 patients due to enlarged corrected QT interval (QTc) at electrocardiogram (ECG); 6 due to unspecified heart disease; 3 due to right bundle branch block; 2 due to changes in ventricular repolarization on the ECG; 1 due to the presence of extrasystoles with irregular heart rhythm; 1 for chronic alcoholism associated with a psychiatric disorder; 3 for old age; 1 due to chronic obstructive pulmonary disease; 4 due to uncontrolled *diabetes mellitus*; 1 due to difficulty in attending the health unit for systemic treatment associated with advanced age; 3 due to liver disease; 6 due to decompensated systemic arterial hypertension; 1 for atrial fibrillation; 1 due to cognitive deficit associated with systemic arterial hypertension; 2 for chronic heart failure and chronic alcoholism; 1 for dilated cardiomyopathy, angina pectoris and chronic renal failure.

[b] One patient due to hypertensive crises; 2 patients due to enlarged QTc at ECG; 1 patient with cardiac arrhythmia; 2 patients due to arthralgia, myalgia and vertigo; 1 patient due to pain at the application site, headache and cyanosis; 1 patient with precordialgia and headache; 1 patient due to myalgia, vertigo and abdominal pain.

[c] One patient due to enlarged QTc at ECG; 1 patient due to left bundle branch block and ECG extrasystoles.

Within ninety days from the beginning of the treatment, 93% of the cases treated with IL MA, 92% of the cases treated with AR and 95% of the cases treated with SR presented epithelialized lesions. At the medical appointment after 120 days from the beginning of the treatment 98% of the cases treated with IL MA, 97% of the cases treated with AR, and 100% of the cases treated with SR presented epithelialized lesions (Fig 5C).

There was no statistical significance for the complete healing time among the groups (p = 0.5) (Fig 5D).

Approximately 75% of the cases treated with IL MA, 75% treated with AR and 79% treated with SR had healed lesions within 180 days of treatment. Within 240 days of treatment, we observed healed lesions in 80%, 90% and 88% of the cases treated with IL MA, AR, and SR respectively.

Kaplan-Meier survival analysis showed that the area of the lesion influenced the epithelialization time. Lesions with an area larger than 900 mm$^2$ took longer to epithelialize (p <0.001) (Fig 5E). Lower limbs injuries took longer to epithelize (p<0.001) (Fig 5F).

There were no statistically significant differences between the rates of epithelialization and complete healing between patients from the state of Rio de Janeiro and other states, although, in survival analysis, patients from Rio de Janeiro presented complete healing of skin lesions earlier than those from other Brazilian regions (p<0.001) (Fig 5G).

The other variables showed an absence of proportionality and there was no statistical significance for epithelialization and complete healing times.

## Etiological identification of *Leishmania* species

Among 592 patients, 468 had a positive culture. Species identification was possible in 405 samples (395 with *L. braziliensis*, five with *L. braziliensis* genetic variant, four with *L.naiffi* and one with *L.naiffi* genetic variant). In the SR group, 36 samples presented *L. braziliensis* (cure rate:33/36). In the AR group, 313 samples presented *L. braziliensis* (cure rate:252/313) and two *L. braziliensis* genetic variant (cure rate:2/2). In the IL group, we identified: 46 samples with *L. braziliensis* (cure rate:27/46), three with *L. braziliensis* genetic variant (cure rate:2/3), four with *L.naiffi* (cure rate:4/4) and one with *L.naiffi* genetic variant (cure rate:1/1).

## Outcomes in subsequent treatments

Ninety-one patients needed a second treatment. Among them, 2 belonged to the SR group, 68 from the AR group and 21 from the IL group. Analyzing the entire cohort, when a second scheme

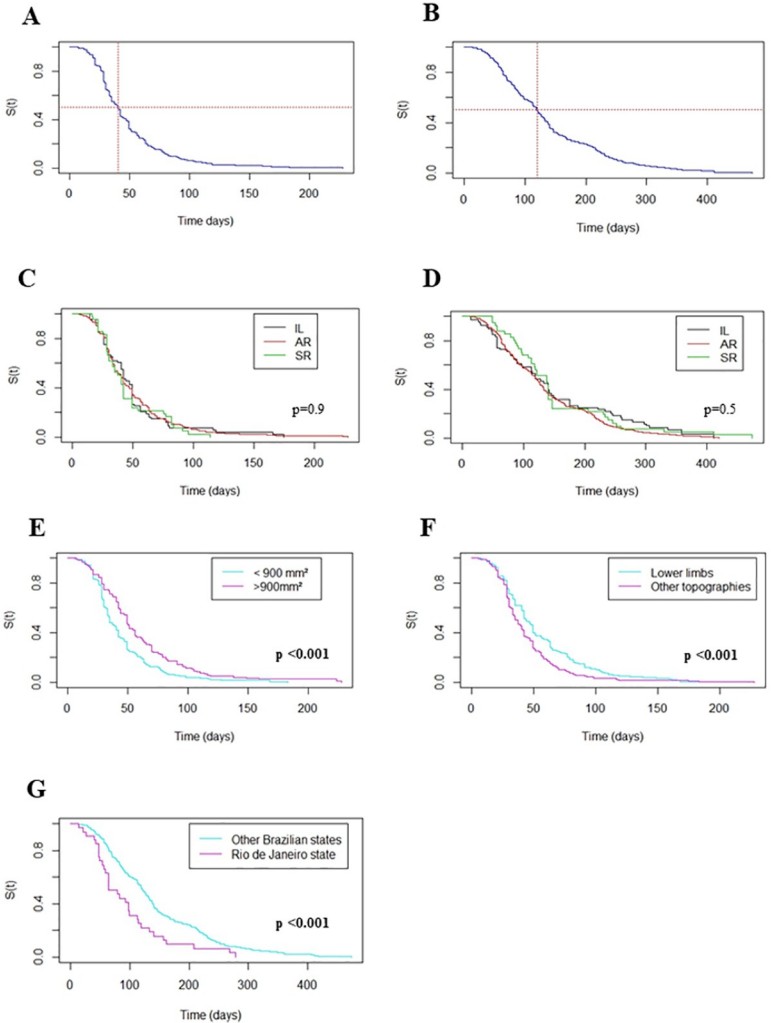

**Fig 5. Kaplan-Meier curves of the analyzed groups and epithelialization / complete healing of cutaneous leishmaniasis lesions.** A) Epithelialization of cutaneous lesions; B) Complete healing of cutaneous lesions; C) Epithelialization of lesions according to the groups; D) Complete healing according to the groups; E) Epithelialization according to lesion´s area; F) Epithelialization according to lesion topography; G) Complete healing according to likely location of infection. SR—standard regimen; AR—alternative regimen; IL—intralesional route; S(t)- Survival function. Bold: p-value <0.05 (according to Log-Rank test).

was necessary (n = 91), the cure rates were: 73.9% (34/46) for AR; 65.6% (21/32) for IL; 100% (5/5) for amphotericin B; 66.7% (4/6) for pentamidine and 100% (1/1) for itraconazole.

In the two cases of the SR group that needed a second treatment, patients received amphotericin B (one) and IL MA (one), achieving therapeutic success of 100% (2/2) (Fig 2). Observing the AR group, when a second treatment was performed (n = 68), the abandonment rate rose to 2.9% and the cure rates were: 72.5% (29/40) for AR; 59.1% (13/22) for IL; 100% (2/2) for amphotericin B; 66.7% (2/3) for pentamidine and 100% (1/1) for itraconazole (Fig 3). In the IL group, in cases of "no cure" in the first treatment (n = 21), the abandonment rate was 4.8% (n = 1) and we verified the cure rates of 83.3% (5/6) for AR; 77.8% (7/9) for IL; 100% (2/2) for amphotericin B and 66.7% (2/3) for pentamidine (Fig 4). For additional information on outcomes, observe data supporting (S1 File).

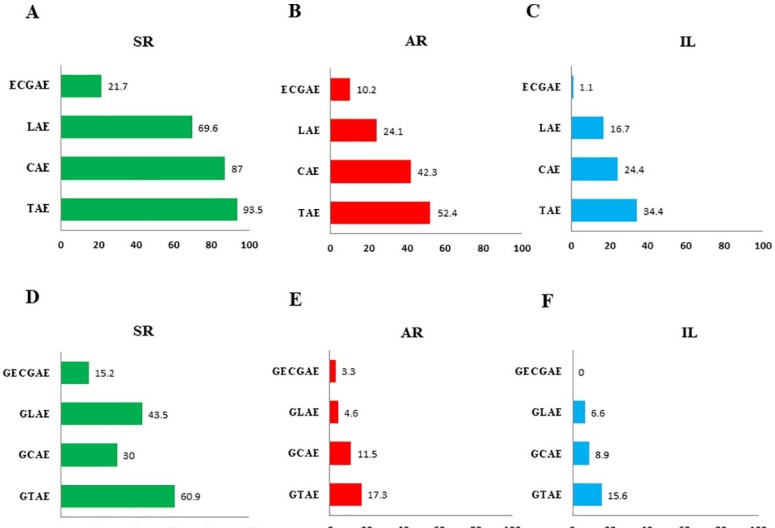

**Fig 6. Adverse events among patients with cutaneous leishmaniasis treated with meglumine antimoniate according to the three groups.** A) Number of cases of SR group with the occurrence of adverse events; B) Number of cases of AR group with the occurrence of adverse events; C) Number of cases of IL group with the occurrence of adverse events; D) Proportion of cases of SR group with moderate to severe adverse events, compared to the other groups; E) Proportion of cases of AR group with moderate to severe adverse events, compared to the other groups; F) Proportion of cases of IL group with moderate to severe adverse events, compared to the other groups. TAE- total adverse events; CAE- clinical adverse events; LAE- laboratory adverse events; ECGAE- Electrocardiographic adverse events; GTAE- grouped (moderate to severe) total adverse events; GCAE- grouped (moderate to severe) clinical adverse events; GLAE- grouped (moderate to severe) laboratory adverse events; GECGAE- grouped (moderate to severe) electrocardiographic adverse events; SR- standard regimen; AR- alternative regimen; IL- intralesional route.

## Toxicity between the groups

The SR group was treated with the median of 25,515 mg of $Sb^{5+}$ cumulative dose (4,860–64,395 mg), 14.4 mL/day. The AR group received the median of 10,004 mg of $Sb^{5+}$ cumulative dose (405–34,020 mg), 4.1 mL/day. The IL group received the median of 1,053 mg of $Sb^{5+}$ cumulative dose (81–8,100 mg), 8 mL/session.

Treatment regimens with systemic MA caused a higher number of AE when compared to the IL group (Fig 6A, 6B and 6C). There was a lower rate of total AE in patients treated with IL (34.4%) and AR (52.4%) groups, comparing to the SR group (93.5%, p< 0.001) (Fig 6A, 6B and 6C).

The SR group presented 30.4% of definitive treatment interruptions due to adverse events. Among these cases, 50% healed despite treatment interruption. Among patients who discontinued treatment due to an adverse event and needed a second treatment, 50% were treated with IL MA and achieved a clinical cure.

When the IL and AR groups were evaluated, the occurrence of total AE, clinical AE and electrocardiographic AE were significantly lower in the IL group (respectively, p = 0.003; 0.002 and 0.003) (Fig 6B and 6C). There were no statistically significant differences regarding laboratory AE between these groups (p = 0.124) (Fig 6B and 6C).

When categorized as moderate to severe AE, there was a clear predominance of these events in the SR group (60.9%; p<0.001) which were related to treatment interruption (Fig 6D, 6E and 6F), as compared to the other groups. AR and IL groups showed no differences between them regarding the occurrence of moderate to severe adverse events [grouped total AE (p = 0.683); grouped clinical AE (p = 0.599); grouped laboratory AE (p = 0.410)] (Fig 6E and 6F).

Among eight cases in the IL group that presented moderate to severe clinical AE, three had local eczema, two had local pain and three presented myalgia and arthralgia. There was no grade 4 (risk to life) AE in the IL group.

## Discussion

Treatment of CL with AR and intralesional infiltrations of MA showed good efficacy and low toxicity as observed in other studies [15,16,20,21,22], even as rescue schemes used after interruption of the treatment due to AE related to other regimens with the same medication.

We found a reasonable complete healing rate in patients treated with AR, with lower abandonment rates. These results were similar to those from the simple blinded, controlled and randomized clinical trial for treatment with systemic MA in different doses and regimens that was performed at INI / Fiocruz, from 2008 to 2013. In this clinical trial, it was observed that intramuscular low doses used in the treatment of CL presented reasonable cure rates and lower toxicity when compared with the standard recommended doses [16].

Patients in the SR group had a higher cure rate than those in the AR group, as previously described [16]. Patients in the SR group had also a higher cure rate than those in the IL group. However, the safety of the AR and IL regimens endorsed their use. The cure rates achieved with AR and IL infiltration were similar to those observed with standard treatments with MA according to other authors in Brazil [20,21] and in the American continent [23].

As stated by recent Brazilian guidelines, patients over the age of 50 should be treated with liposomal amphotericin B [8]. We highlight the high number of patients who were safely treated with AR, avoiding the inconveniences of using liposomal amphotericin B (costs with hospitalization, daily monitoring with laboratory tests, absence from work and family), which in certain circumstances make liposomal amphotericin B an unfeasible treatment, especially in primary health care units. Many CL patients live in small cities or rural settlements far from the reference treatment centers, with great difficulties in the adherence to liposomal amphotericin B treatment. Notably, 128 patients (21.6% of the cohort) aged over 50 years and therefore with a formal indication for the use of liposomal amphotericin B according to the recent Brazilian consensus, were successfully treated with AR and IL.

Besides, the AR group had a large number of patients with formal contraindications to systemic therapy. Even so, we highlight that they concluded AR treatment with a high rate of effectiveness, ten times fewer interruptions and three times fewer abandonments than in the SR group, which corroborates other studies [9,11,15,16,24].

Despite the effectiveness of SR being significantly higher than AR and IL, we emphasize that the complete cure rate between the AR and the IL groups did not show a statistical difference after the first treatment. An impressive number of patients with contraindications to systemic MA (45.6% of the cohort) who would be treated with liposomal amphotericin B according to formal indication were treated with AR or IL regimens, with safety and practicality.

Adverse events, including clinical, laboratory and electrocardiographic events, were more common in patients treated with SR, a fact that corroborates the higher safety of the AR and IL route; when AE occurred in these groups, mild AE were the most common. AE may occur in both intramuscular and IL treatment, but in the latter, they are usually milder [8,25,26]. However, IL MA should be used cautiously when previous cutaneous eczema is detected [26]. Besides, even though there were more patients with comorbidities in the IL group, patients had a reduced rate of AE, which demonstrates the safety of the IL route.

Half of the patients in the SR group who stopped treatment after AE healed without any additional therapeutic approach. It is inferred that the use of MA causes improvement even

after the drug is discontinued. This finding supports the treatment with MA in series of ten days interspersed with a ten-day interval in selected cases [12]. In the SR group, among the cases that discontinued treatment due to AE and required a second therapeutic regimen, half received successful IL infiltrations. These patients had a formal indication for treatment with other drugs, but they were treated with an IL MA regimen as a rescue and safe therapy.

Historically, Oliveira-Neto and collaborators described the success of IL MA in the New World cutaneous leishmaniasis in 1997, highlighting the practicality and reduced toxicity of this strategy, even in fieldwork [10,24]. When publishing the more recent guideline for the treatment of CL, the Brazilian Ministry of Health recommended this technique as it was standardized at our reference center [6,8], because of the following characteristics: easy acquisition of the compound, low cost, ease of replication of the technique, reduced need for training of specialized professionals, need of simpler laboratory resources during follow-up and finally the great expertise of the group at INI/Fiocruz in successfully carrying out the technique for more than 30 years. This technique may be safely used even when infrastructure is poor [21,27]. Brazil is a country with continental dimensions and this treatment option could be easily and safely performed in primary health care units [21] even in remote areas, such as rural settlements or indigenous villages.

In cases with moderate to severe AE that led to the interruption of the intramuscular regimens, good adherence to the IL route and clinical cure were observed. When using the IL route, in general, a low abandonment rate was noted, with greater adherence to treatment. The IL group showed more comorbidities and consequently larger use of concomitant medications to treat them, reinforcing its historical use for this purpose at INI. Also, patients who received IL treatment were often elderly people who frequently used other drugs to treat comorbidities (which commonly represented contraindications to intramuscular or intravenous MA) [11,12].

Most cases in the IL group were treated with two sessions, although there was great variation in the number of sessions until a clinical cure was achieved. The ideal number of infiltrations remains a challenge, especially considering that this number affects the effectiveness of the treatment [21]. There are great variations regarding the number of infiltrations carried out in different studies [21,27]. Standardization of the number of infiltrations is essential to better evaluate the effectiveness of this treatment. However, in patients with contraindications to intravenous or intramuscular MA, a greater number of infiltrations may still represent a beneficial strategy. Patients unsuccessfully treated with two doses of IL MA might have evolved with a complete healing outcome if a greater number of infiltrations had been performed, since some patients with failures to the initial IL regimen were treated again after variable periods with the same therapeutic modality and achieved cure. Even with multiple infiltrations, the cumulative dose of MA in the IL route is lower than that used through intravenous or intramuscular therapy. After treatment, clinical evaluation of the progression of the epithelialization process and its subsequent steps towards complete healing every two weeks is essential to the indication of new infiltrations, as pointed out by other authors [21,27].

In the survival analysis, the median time of epithelialization and complete healing was similar in the three analyzed groups. The healing process involves the development of a competent immune response that often does not depend on the treatment route, since the balance between type 1 and type 2 immune responses seems to be a determining factor for clinical forms of benign evolution or even spontaneous resolution [28,29,30,31]. The lesion evolution time greater than two months in the IL group may have led to the establishment of a more complete immune response, favouring natural infection control, with a lower parasitic burden [32,33].

Drugs with leishmanicidal properties when topically applied must be able to cross the corneal barrier of the skin, whose physicochemical properties hinder the penetration and local concentration of the drug. On the other side, drugs for IL treatment must penetrate in deeper layers of the dermis and reach parasitized macrophages, and the effectiveness of the treatment depends on this characteristic and the peculiarities of the infectious species [34].

The increased rates of epithelialization and complete healing with longer follow-up (up to 180 days or more) suggest that an expectant post-treatment conduct with periodic (every two weeks) evaluations of the patients until epithelialization when the lesion is showing progressive signals of improvement is often more advisable than premature re-treatments [16,33].

Survival analysis shows that the size and location of the lesion influence the epithelialization time in CL, although the complete healing time was not influenced by these factors [35], as noted by other authors [36]. Individuals from other Brazilian states than Rio de Janeiro had longer healing times, which might be correlated with a delay in diagnosis and difficulties in accessing the health unit for follow-up, a higher rate of absences and less adherence to the proposed therapy. However, there was no difference between the patients regarding the likely local of infection (from the state of Rio de Janeiro or not) in proportions of complete healing. In the analysis of epithelialization rates, there was also no difference among the groups, corroborating the observations of other authors [14].

Regarding the shorter healing time of the lesions in the group treated with intramuscular MA, the number of moderate to severe AE with increased morbidity not related to the disease itself and the more frequent interruption of the treatment are relevant points. In comparison with the IL route and AR, a greater number of abandonments and less adherence to therapy was observed with the use of SR.

A small retrospective study using a non-standard IL technique demonstrated a complete healing rate of 67.7% [20]. The same authors tried to validate a standard operational procedure for IL infiltration of MA but ended up concluding that the methodology developed at our institute and certified by the BMH is reproducible and may be used by health professionals with minimal training with success and safety [37]. In 2018, the same group published a single-arm phase II clinical trial with 53 patients treated with IL MA, with a cure rate of 87% and a median of 7 infiltrations [38]. Hypersensitivity reactions were more common than those observed in other studies [12,39], perhaps due to a larger number of IL infiltrations of the drug. Rodrigues and colleagues recently found an efficacy rate of 66.7% for IL treatment in a retrospective small cohort of 21 cases [22], this variation of effectiveness may be associated with the technique of application, personal variables of the patients and even the circulating strain. However, most patients from other Brazilian states included in this and other studies have been successfully treated with AR or IL [14,21,22,36]. In the nearby state of Minas Gerais, the effectiveness of the IL treatment of CL was 66.7% [21,22].

In the context of Latin America, ninety-two patients were treated through intralesional route with an antimonial lidocaine- MA compound with good effectiveness and a low number of side effects [40], however, we argue that it is more advantageous to use the standard drug with no anesthetic, due to lesser risk of contamination of the standard drug. IL MA was described as a safe and effective option treatment for CL in a Bolivian rainforest rural area [41].

Despite the use of a formula for calculating the area of the lesion that differs from the Pan American Health Organization [42], the cutoff point of 900 mm$^2$ for the area was used to standardize the comparison among different literature data. We emphasize the efficiency of MA infiltrations in lesions with areas greater than 900mm$^2$ and in those situated in topography where greater expertise from the medical personnel is required (including head and periarticular areas). The mastery of the infiltration technique and a good knowledge of local anatomy

turn it safe for the treatment of any corporal segment [6,13]. Even though there were no differences between the treatment groups and the topography (cephalic segment and periarticular regions) of the cutaneous lesion, as found in several studies [13,21,22,24,27,36], it is recommended to perform controlled prospective studies.

Despite AR and IL treatments have been used predominantly in the Brazilian state of Rio de Janeiro, several authors showed reasonable effectiveness of these schemes when compared to SR [10,11,14,16,21,22], with the advantage of lower morbidity. Even though a species-specific variation sensitivity in different areas of the country is expected, MA resistance is not a public health problem in Brazil [7,8,16]. Differences have not been described in therapeutic response among patients from different Brazilian states treated with MA [8,14,15,21]. Different circulating strains of *L. (V.) braziliensis* in Brazilian territory do not appear to show resistance to pentavalent antimonials, however the circulating strain in the state of Rio de Janeiro proved to be sensitive to MA [43]. There are no clinical trials with AR in other Brazilian states than Rio de Janeiro, but in our previous low-dose reports, we usually include around 15% of patients from other states successfully treated [10,14,15]. This allows us to raise the hypothesis that low-dose could be used especially when patient safety is a primary concern [8,16].

Although the genetic variability of *L.braziliensis* in the state of Rio de Janeiro is lower than in other regions of the country [44,45], it seems that patients residing in Rio de Janeiro are not more susceptible to treatment with MA, when compared to patients from other Brazilian states [14]. Besides, a clinical trial with the alternative MA regime was performed only in Rio de Janeiro [16]. The present study did not show any difference in the effectiveness of treatments performed on patients from Rio de Janeiro or those from other Brazilian states.

IL MA has been widely used in other states and countries in Latin America with proven efficacy [7,11,20,21,22,27,40,41]. A multicenter clinical trial with IL MA is being concluded in Brazil with the coordination of our reference center.

Although this cohort has comprised mostly cases from the state of Rio de Janeiro, Brazil, the high number of patients was relevant. This reference center also receives patients from various states in the country and has a multidisciplinary team and physical and laboratory infrastructure for monitoring the patients during and after treatment. This study demonstrates the robustness of a referral center, with patients followed up for many years after treatment with the so-called alternative schemes (AR and IL) and its results can be extrapolated to other centers with similar conditions.

There was a wide variation in the number of MA infiltration sessions, although the effectiveness of this therapeutic modality was similar to others studies [11,12,21,22,37].

This study has limitations that should be discussed. The first point is the observational study design with descriptive character that represents the reality of clinical practice in a large reference center, with an expressive number of patients over a long period, where patients are mostly treated with AR. In cases where there was a formal contraindication to systemic MA, and when other medications are indicated but are not tolerable (amphotericin B, for example), IL treatment has been traditionally used. Patients treated with SR, in general, were part of research projects. In this context, there was a marked numerical difference among the number of subjects in each group, representing a limitation of the cohort. When comparing the groups using proportions, the relevance of the different size of each group can be minimized.

The second limiting point was the data loss, typical limitation of retrospective studies that analyze such a long period. This was a retrospective study in which there was occasionally missing data related to patient follow-up and adverse events, mainly the mild ones that may not be reported in the clinical evaluations because they are not thoroughly searched. Even though, we are a national reference center in ATL treatment, with serial clinical and laboratory evaluation of the patients.

The third limitation concerns to the extrapolation of these findings to other cohorts (external validation). We present a cohort of a reference center and as such, inference is only possible for the same type of population.

Factors that favour reduced abandonment rates at INI/Fiocruz included: establishment of a good doctor-patient relationship; protocols for contacting the absent patients (phone calls, emails, contact with close relatives, among others); and transportation frequently provided by the public health system in Brazil that guarantees the return of the patients, with no additional costs for them.

As a reference center for leishmaniasis, many patients are allocated in research projects, which favour the participants to understand their involvement in the treatment. The careful evaluation of the multidisciplinary team and the occasional availability of resources for transportation and feeding, as well as the pro-activity from the health professionals in contacting missing patients, are expected to minimize follow-up losses.

All the studied MA regimens proved to be effective for the treatment of CL, but AR and IL regimens were less toxic and might be chosen as first-line treatments especially in older people with comorbidities using other non-specific medications. Both regimens showed similar efficacy in epithelialization and complete healing of lesions when compared with those observed with SR, with the advantage of a lower number and milder adverse events that rarely imply interruption of treatment. Thus, AR and IL treatments were effective and safe ways to treat CL.

IL MA was effective and associated with low toxicity, even in cases where it had been used as a rescue therapy after unsuccessful treatment with SR and AR systemic MA. Besides, the IL route was successfully used in lesions with diameters larger than 3 cm and those located in periarticular areas and head.

We present a cohort of a reference center which represents a limitation but at the same time a strong point of the study. Further studies using standardized application technique, fixed number of infiltrations, follow-up on pre-set days and assessment of adverse events and outcomes are underway at INI/Fiocruz.

## Supporting information

**S1 Fig. Flowchart of inclusion and exclusion of the studied patients, for the population of patients with American tegumentary leishmaniasis treated at INI, 2000–2017.** MA- meglumine antimoniate; INI- Evandro Chagas National Institute of Infectious Diseases; Fiocruz- Oswaldo Cruz Foundation; ML- mucosal leishmaniasis; MCL- mucocutaneous leishmaniasis; ESH- early spontaneous resolution without treatment; DCL- disseminated cutaneous leishmaniasis; CL- cutaneous leishmaniasis; HIV- human immunodeficiency virus; LCL- localized cutaneous leishmaniasis; SR- standard regimen with MA 10 to 20 mg / kg / day; AR- alternative regimen with MA 5 mg / kg / day; IL- intralesional MA. *MA used as first therapeutic regimen.
(TIF)

**S2 Fig. Patients follow-up: standard regimen group.** SR- standard regimen; IL- intralesional route.
(TIF)

**S3 Fig. Patients follow-up: alternative regimen group.** AR- alternative regimen; SR-standard regimen; IL- intralesional route; AE—adverse events.
(TIF)

**S4 Fig. Patients follow-up: intralesional route.** IL- intralesional route; AR- alternative regimen; SR- standard regimen.
(TIF)

**S5 Fig. Kaplan-Meier curves of the analyzed groups and epithelialization /complete healing of cutaneous leishmaniasis lesions.** A) Epithelialization of cutaneous lesions; B) Complete healing of cutaneous lesions; C) Epithelialization of lesions according to the groups; D) Complete healing according to the groups; E) Epithelialization according to lesion´s area; F) Epithelialization according to lesion topography; G) Complete healing according to likely location of infection. SR- standard regimen; AR- alternative regimen; IL- intralesional route; S(t)- Survival function. Bold: p-value <0.05 (according to Log-Rank test).
(TIF)

**S6 Fig. Adverse events among patients with cutaneous leishmaniasis treated with meglumine antimoniate according to the three groups.** A) Number of cases of SR group with the occurrence of adverse events; B) Number of cases of AR group with the occurrence of adverse events; C) Number of cases of IL group with the occurrence of adverse events; D) Proportion of cases of SR group with moderate to severe adverse events, compared to the other groups; E) Proportion of cases of AR group with moderate to severe adverse events, compared to the other groups; F) Proportion of cases of IL group with moderate to severe adverse events, compared to the other groups. TAE- total adverse events; CAE- clinical adverse events; LAE- laboratory adverse events; ECGAE- Electrocardiographic adverse events; GTAE- grouped (moderate to severe) total adverse events; GCAE- grouped (moderate to severe) clinical adverse events; GLAE- grouped (moderate to severe) laboratory adverse events; GECGAE- grouped (moderate to severe) electrocardiographic adverse events; SR- standard regimen; AR- alternative regimen; IL- intralesional route.
(TIF)

**S1 File. Outcomes in subsequent treatments.** SR- standard regimen; AR- alternative regimen; IL- intralesional route.
(DOCX)

## Acknowledgments

We thank the Evandro Chagas National Institute of Infectious Diseases (INI / Fiocruz) for the use of the infrastructure.

## Author Contributions

**Conceptualization:** Carla Oliveira-Ribeiro, Maria Inês Fernandes Pimentel, Liliane de Fátima Antonio Oliveira, Érica de Camargo Ferreira e Vasconcellos, Fatima Conceição-Silva, Armando de Oliveira Schubach, Luciana de Freitas Campos Miranda, Ana Cristina da Costa Martins, Raquel de Vasconcellos Carvalhaes de Oliveira, Leonardo Pereira Quintella, Marcelo Rosandiski Lyra.

**Data curation:** Maria Inês Fernandes Pimentel, Liliane de Fátima Antonio Oliveira, Érica de Camargo Ferreira e Vasconcellos, Eliame Mouta-Confort, Claudia Maria Valete-Rosalino, Marcelo Rosandiski Lyra.

**Formal analysis:** Carla Oliveira-Ribeiro, Liliane de Fátima Antonio Oliveira, Fatima Conceição-Silva, Cintia Xavier de Mello, Raquel de Vasconcellos Carvalhaes de Oliveira, Marcelo Rosandiski Lyra.

**Investigation:** Carla Oliveira-Ribeiro, Armando de Oliveira Schubach, Aline Fagundes, Cintia Xavier de Mello, Eliame Mouta-Confort, Ana Cristina da Costa Martins, Marcelo Rosandiski Lyra.

**Methodology:** Carla Oliveira-Ribeiro, Maria Inês Fernandes Pimentel, Liliane de Fátima Antonio Oliveira, Érica de Camargo Ferreira e Vasconcellos, Fatima Conceição-Silva, Armando de Oliveira Schubach, Aline Fagundes, Cintia Xavier de Mello, Luciana de Freitas Campos Miranda, Raquel de Vasconcellos Carvalhaes de Oliveira, Leonardo Pereira Quintella, Marcelo Rosandiski Lyra.

**Project administration:** Érica de Camargo Ferreira e Vasconcellos, Marcelo Rosandiski Lyra.

**Software:** Luciana de Freitas Campos Miranda, Raquel de Vasconcellos Carvalhaes de Oliveira.

**Supervision:** Carla Oliveira-Ribeiro, Maria Inês Fernandes Pimentel, Érica de Camargo Ferreira e Vasconcellos, Armando de Oliveira Schubach, Cintia Xavier de Mello, Ana Cristina da Costa Martins, Raquel de Vasconcellos Carvalhaes de Oliveira, Leonardo Pereira Quintella, Marcelo Rosandiski Lyra.

**Validation:** Carla Oliveira-Ribeiro, Maria Inês Fernandes Pimentel, Armando de Oliveira Schubach, Luciana de Freitas Campos Miranda, Claudia Maria Valete-Rosalino, Ana Cristina da Costa Martins, Leonardo Pereira Quintella, Marcelo Rosandiski Lyra.

**Writing – original draft:** Carla Oliveira-Ribeiro, Maria Inês Fernandes Pimentel, Liliane de Fátima Antonio Oliveira, Érica de Camargo Ferreira e Vasconcellos, Fatima Conceição-Silva, Armando de Oliveira Schubach, Aline Fagundes, Cintia Xavier de Mello, Eliame Mouta-Confort, Luciana de Freitas Campos Miranda, Claudia Maria Valete-Rosalino, Ana Cristina da Costa Martins, Raquel de Vasconcellos Carvalhaes de Oliveira, Leonardo Pereira Quintella, Marcelo Rosandiski Lyra.

**Writing – review & editing:** Carla Oliveira-Ribeiro, Maria Inês Fernandes Pimentel, Liliane de Fátima Antonio Oliveira, Érica de Camargo Ferreira e Vasconcellos, Fatima Conceição-Silva, Armando de Oliveira Schubach, Aline Fagundes, Cintia Xavier de Mello, Claudia Maria Valete-Rosalino, Raquel de Vasconcellos Carvalhaes de Oliveira, Marcelo Rosandiski Lyra.

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
