## [Decision Letter · Decision Letter 0]

28 Feb 2021

Dear Dr Ribeiro,

Thank you very much for submitting your manuscript "An old drug and different ways to treat cutaneous leishmaniasis: intralesional and intramuscular meglumine antimoniate in a reference center, Rio de Janeiro, Brazil" for consideration at PLOS Neglected Tropical Diseases. As with all papers reviewed by the journal, your manuscript was reviewed by members of the editorial board and by several independent reviewers. In light of the reviews (below this email), we would like to invite the resubmission of a significantly-revised version that takes into account the reviewers' comments. 

We cannot make any decision about publication until we have seen the revised manuscript and your response to the reviewers' comments. Your revised manuscript is also likely to be sent to reviewers for further evaluation.

Sincerely,

Gregory Deye

Associate Editor

Helen Price

Deputy Editor

Reviewer's Responses to Questions

**Key Review Criteria Required for Acceptance?**

**Methods**

-Are the objectives of the study clearly articulated with a clear testable hypothesis stated?

-Is the study design appropriate to address the stated objectives?

-Is the population clearly described and appropriate for the hypothesis being tested?

-Is the sample size sufficient to ensure adequate power to address the hypothesis being tested?

-Were correct statistical analysis used to support conclusions?

-Are there concerns about ethical or regulatory requirements being met?

Reviewer #1: The objectives of this study are clearly defined, however there is no power calculation discussed to determine if the groups in each arm meets statistical significance. The populations is defined except for the infecting species of Leishmania. No ethical concerns noted. Under exclusion criteria it mentions excluding for prior CL treatment, but does not give a time frame for this.

Reviewer #2: There is no clear statement of the hypotheses of the study in the document.

Because it is a retrospective study, there are marked differences in the size of the groups under evaluation, which potentially generates biases.

There are no ethical problems.

**Results**

-Does the analysis presented match the analysis plan?

-Are the results clearly and completely presented?

-Are the figures (Tables, Images) of sufficient quality for clarity?

Reviewer #1: There are over 100 data points missing on several variables and the size of each arm is vastly different. Figure 1 and the text describing it do not match.

Reviewer #2: The analysis of the information is adequate but the results are only presented as percentages, without showing confidence intervals that allow the reader to better understand the findings.

The authors should improve the clarity of the images in figure 6.

**Conclusions**

-Are the conclusions supported by the data presented?

-Are the limitations of analysis clearly described?

-Do the authors discuss how these data can be helpful to advance our understanding of the topic under study?

-Is public health relevance addressed?

Reviewer #1: Given the size discrepancies between the arms it is hard to draw conclusions, however it clearly seems that the standard regimen of MA is poorly tolerated and the other regimens are likely good alternatives.

Reviewer #2: Yes partially. This is a retrospective observational study with limitations in its external validity, therefore its conclusions should be referential. The authors highlight that the use of low-dose MA in the alternative (AR) and intralesional regimens is effective and has a significantly lower toxicity than the standard regimen (recommended in the Brazilian guidelines for the treatment of LC), but they do not mention that the regimen SR with MA was statistically better than the AR and IL regimens.

The authors should discuss the peculiar situation of the Leishmania strain that circulates in the State of RJ is sensitive to low doses of MA. There is information in the Brazilian literature that explains this type of response.

Authors should describe the limitations of the study.

**Editorial and Data Presentation Modifications?**

Reviewer #1: (No Response)

Reviewer #2: No

**Summary and General Comments**

Reviewer #1: This is a single-center, retrospective study or 592 patients with parasitologically diagnosed CL evaluating standard regimen MA, alternative regimen MA, and intralesional MA. The main findings are that all 3 arms have the same sure rate, but the standard regimen has more side effects and abandonment than the other arms. Given its retrospective nature the arms are not equal and the standard arm has only 46 subjects comparted to the alternative arm with 456, and the IL arm with 90. This makes it difficult to adequately compare the arms. Additionally variables are not available for many subjects with over 100 points missing on more than one variable used in the analysis. 

One major issue for the paper is that it discusses that the IL subjects return for more than one treatment, however the number of treatments and the average amount of MA used for each treatment is not discussed. It is fundamental to understand the amount of drug used to understand how AEs come into play in this type of study. Also, a major concern with giving local vs systemic treatment is the development of ML. They do mention that there is an initial evaluation for ML and those are excluded, but does not provide follow up to show if mucosal lesions developed. Also, the results say that IL has no treatment interruption, but how can this be defined when there is no pre-set schedule for giving the IL treatments?

Minor issues are Figure 1 not showing similar items to what is discussed in the text. It is unclear why there are 47 subjects excluded from the study, where they not treated in the study timeframe? Line 135 refers to 10 subjects being excluded which is not reflected in the figure. Exclusion criteria do not match what is written in the text. Also, using more commonly used English terms like 2 weeks instead of fortnight.

Reviewer #2: This study the authors evaluate the effectiveness and toxicity to meglumine antominiate (MA) in 592 naive patients with cutaneous lesihmaniasis (CL) with parasitological diagnosis treated with MA as the first therapeutic option in INI / Fiocruz, State of Rio Janeiro (RJ), Brazil, from January 2000 to December 2017. In this retrospective cohort, 3 treatment regimens with MA were used: Alternative Regime (AR), Standard Regime (SR) and Intralesional therapy (IL). SR is recommended in the Brazilian guidelines for the treatment of CL.

Several publications in RJ have demonstrated the efficacy of IL therapy with MA in LC and of low doses of MA, although significantly lower CR than the SR regimen (Saheki et al 2017, ref # 16). The greatest value of this study is to show as an alternative the AR regimen, traditionally used in INI / Fiocruz in RJ, with a cure rate of 84.3% versus 95.3% of SR, a lower effectiveness but with the advantage of being 10 times less toxic and have 3 times less abandons. This is a reasonable rate, especially for patients over 50 years of age, who have contraindications to the use of pentavalent antimonials due to the frequency and severity of adverse effects.

This cohort has several peculiarities: (i) in INI / Fiocruz (ATL referral center) the treatment regimen of choice is AR (77% of the patients evaluated), (ii) the SR (7.8%) and IL regimens ( 15.2%) are regularly used in LC patients with contraindications to the use of pentavalent antimonials by the systemic route (page 5, lines 113-177), (iii) for the AR regimen they use low doses of MA (5mg Sb5+/Kg/day) for 30 consecutive days, (iv) the majority of 552/592 patients (93.4%) were infected in the State of Rio de Janeiro, (v) they have a surprising low abandonment rate at 3 (2.4%) and 6 (%) months , very different from the experience of other hospitals where the dropout rate is greater than 50% in this time (the researchers should make a brief description of the success of this retention).

Limitations

1. The 3 treatment groups are not comparable, by nature how the treatment regimen was selected. Therefore, the reported effectiveness does not show superiority or inferiority of any of the 3 regimens used and the information is only referential.

2. Due to the way in which patients have been managed (frequently MA has continued to be used between 2 to 4 times) without the authors having defined criteria for repeating or changing the regimen, it makes the effectiveness rates not comparable and difficult to interpret from the second scheme used.

3. It is suggested not to use the term treatment failure, because in practice they have used the “no cure” criterion, which explains continuing to use the same drug (MA), although with a different regimen.

4. Due to the nature of the study (retrospective observational), it has poor external validity that limits the extrapolation of conclusions for all of Brazil. The authors report that there are no statistical differences in the re-epithelialization rate (cure criterion at 3 months) between patients from Rio de Janeiro and other Brazilian states (lines 232-234), implicitly suggesting that the therapeutic response is similar. However, there is a clear contradiction with the results that the authors show in Figure 3, Picture G, where the healing rate (definitive cure criterion at 6 months) in is lower (p <0.001) in lesions acquired in other different states RJ. Furthermore, there are several reports in the literature with variations in the efficacy of Sb5 + in Leishmania (V.) braziliensis in different endemic areas of Brazil. Apparently the L. (V.) braziliensis strain circulating in RJ is susceptible to low doses of pentavalent antimonials. The authors do not comment on this evidence.

5. The cure rates (CR), only presented as percentages, should include the numerators and denominators, as well as the confidence intervals so that the reader can understand this complex study. Thus, the authors with the first AM scheme in the RA regimen report a CR of 84.3% and then mention that the cure rate for retreatment (understood as RA) was 69.2% (68 cases). Analyzing figure 3, only 56 patients received RA retreatment and a decrease in CR is observed with the 2nd treatment scheme 28/39 (71.8%), 3rd treatment 7/12 (53.8%) and 4th treatment 2/5 (40%). This message is important and not all treatment regimens should be grouped together because they have variable cure rates depending on the number of retreatments. It is also not advisable to mix IL regimens with RA because the CRs can vary. For this reason, the authors' description of retreatment (page 15) is confusing and does not have a clear message. Same comment for description of IL results.

6. The authors report epithelialization and healing time with no statistical difference between the 3 groups (lines 267-269), however, as they have different interventions (some more than one treatment scheme) the information is only referential. Based on these findings, the authors should recommend that prospective clinical studies be conducted.

Authors should have a study limitations section. This study has important limitations as it is a retrospective observational study.

The conclusion must be corrected, the effectiveness of the standard regimen with MA was statistically better than the AR and IL regimens, although with significantly less toxicity, an aspect that is not mentioned in the conclusion of the study abstract.

PLOS authors have the option to publish the peer review history of their article (what does this mean?). If published, this will include your full peer review and any attached files.

Reviewer #1: No

Reviewer #2: No
---

## [Editor Report · Decision Letter 1]

30 Apr 2021

Dear Dr Ribeiro,

Thank you very much for submitting your manuscript "An old drug and different ways to treat cutaneous leishmaniasis: intralesional and intramuscular meglumine antimoniate in a reference center, Rio de Janeiro, Brazil" for consideration at PLOS Neglected Tropical Diseases. As with all papers reviewed by the journal, your manuscript was reviewed by members of the editorial board and by several independent reviewers. In light of the reviews (below this email), we would like to invite the resubmission of a significantly-revised version that takes into account the reviewers' comments. 

Thank you for your submission of your revised manuscript. After careful review of your revision, it is clear that many of the most substantive critiques by the reviewers to the orignial manuscript have not been adequately incorporated into the revised manuscript. Among these is a more thorough discussion of study limitations, the potential impact of sensitivity to MA on external validity, and a more careful approach to comparative statements in recognition of the observational nature of this study. I agree that incorporating the critiques of the prior reviewers, rather than arguing against them would strengthen the manuscript. I do hope that you will make these revisions and resubmit because I think that sharing such a large treatment experience would have value to the treating community and would make a valuable addition to the published literature on the subject.

We cannot make any decision about publication until we have seen the revised manuscript and your response to the reviewers' comments. Your revised manuscript is also likely to be sent to reviewers for further evaluation.

Sincerely,

Gregory Deye

Associate Editor

Helen Price

Deputy Editor

Thank you for your submission of your revised manuscript. After careful review of your revision, it is clear that many of the most substantive critiques by the reviewers to the orignial manuscript have not been adequately incorporated into the revised manuscript. Among these is a more thorough discussion of study limitations, the potential impact of sensitivity to MA on external validity, and a more careful approach to comparative statements in recognition of the observational nature of this study. I agree that incorporating the critiques of the prior reviewers, rather than arguing against them would strengthen the manuscript. I do hope that you will make these revisions and resubmit because I think that sharing such a large treatment experience would have value to the treating community and would make a valuable addition to the published literature on the subject.
---

## [Editor Report · Decision Letter 2]

16 Aug 2021

Dear Dr Ribeiro,

We are pleased to inform you that your manuscript 'An old drug and different ways to treat cutaneous leishmaniasis: intralesional and intramuscular meglumine antimoniate in a reference center, Rio de Janeiro, Brazil' has been provisionally accepted for publication in PLOS Neglected Tropical Diseases.

Best regards,

Gregory Deye

Associate Editor

Helen Price

Deputy Editor

This manuscript is ready for publication, however it still needs some English copyediting. Please see the attached document with recommended changes and incorporate them during the final technical checks.

---

## [Editor Report · Acceptance letter]

15 Sep 2021

Dear Dr Oliveira-Ribeiro,

We are delighted to inform you that your manuscript, "An old drug and different ways to treat cutaneous leishmaniasis: intralesional and intramuscular meglumine antimoniate in a reference center, Rio de Janeiro, Brazil," has been formally accepted for publication in PLOS Neglected Tropical Diseases.

Best regards,

Shaden Kamhawi

co-Editor-in-Chief

Paul Brindley

co-Editor-in-Chief
